# Digital Subtraction Pulmonary Angiography in Children with Pulmonary Hypertension due to Bronchopulmonary Dysplasia

**DOI:** 10.3390/medsci7020026

**Published:** 2019-02-08

**Authors:** Bibhuti Das, Michelle-Marie Jadotte, Jaime Mills, Kak-Chen Chan

**Affiliations:** Joe DiMaggio Children’s Hospital Heart Institute, Memorial Healthcare System, Hollywood, FL 33021, USA; MJadotte@mhs.net (M.-M.J.); jaimenmills@gmail.com (J.M.); KCChan@mhs.net (K.-C.C.)

**Keywords:** bronchopulmonary dysplasia, pulmonary hypertension, preterm infants

## Abstract

Bronchopulmonary dysplasia (BPD) is the most common respiratory sequelae of prematurity and histopathologically features fewer, dysmorphic, pulmonary arteries. We present our experience with the digital subtraction pulmonary angiography (DSPA) findings of a segmental vascular filling abnormality in three children who were born at extreme prematurity and have pulmonary hypertension due to severe BPD. Our preliminary data suggest that DSPA may be useful in evaluating the severity of pulmonary vascular disease in children with BPD.

## 1. Introduction

Bronchopulmonary dysplasia (BPD) is a chronic lung disease of primarily premature infants that results from an imbalance between lung injury and repair in developing and immature lungs [1]. The overall incidence of BPD at a gestational age of 28 weeks is high (23%), but progressively declines in infants 28–32 weeks of age (10.6%) [2]. Infants with BPD are at increased risk for pulmonary hypertension (PH), with right ventricular failure or chronic lung morbidities contributing to poor outcomes. Data from neonatal rats have shown that chronic hypoxia during the first two weeks of life (a period of lung development corresponding to human fetal lung development during the third trimester) has led to interference in alveolar and pulmonary angiogenesis [3]. Histopathological changes include alveolar simplification and dysmorphic pulmonary vascularization. Human infants who die from BPD have abnormal alveolar micro vessels, but such changes in the pulmonary vasculature in preterm infants have not been shown by pulmonary angiography.

Digital subtraction angiography is an old technique, integrating digital data collection and computer processing to produce a medical image. The diagnostic role of digital subtraction angiography in patients with pulmonary embolism, and with congenital anomalies of pulmonary vessels, has been well established [4]. Using digital subtraction pulmonary angiography (DSPA), the present retrospective study aims to characterize abnormalities in the pulmonary vasculature in three children who were born at extreme prematurity and have PH due to severe BPD.

## 2. Methods

The present study was part of a pilot project to evaluate pulmonary vasculature during the cardiac catheterization for PH. This study included three infants who were between 22 and 24 completed weeks of gestation at birth. The right heart catheterization was performed to define the hemodynamic profile, to test vasoreactivity, and to formulate specific treatment plans for PH management, as their echocardiogram showed evidence of persistently high right ventricular pressure despite a combination of pulmonary vasodilator therapies. 

This retrospective pilot study was approved by the Memorial Healthcare System Institutional Review Board (MHS.2018.116). Informed written parental consent was obtained from parents for cardiac catheterization, and the study adhered to the standards set by the latest version of the Declaration of Helisinki. Retrospective medical chart reviews were performed to collect the following variables: age, sex, gestational age, birth weight, duration and type of respiratory support after birth, echocardiographic findings and PH therapy prior to cardiac catheterization, and hemodynamic data at the right heart catheterization. The right heart catheterization was performed as per the standard of care for each patient. A digital subtraction pulmonary artery angiogram was performed in each patient instead of a standard pulmonary angiogram because of the potential advantages of both a lower radiation exposure and a lower dose of radiographic contrast material compared to conventional cineangiography. All patients tolerated the procedure without any complications.

Our DSPA technique used digital images to perform “instant” subtraction, contrast enhancement, and pixel shifting [5]. The first image (the mask) was obtained without the contrast; successive images were obtained after injecting 1 ml/kg of the contrast (1:2 diluted) through an angiographic catheter in the pulmonary artery, and then images were acquired at a rate of 15 frames/sec. Respiration was suspended for the duration of the angiogram. The mask image was digitally subtracted from the successive contrast images automatically using dynamic masking and the time-interval-difference mode, which rendered the contrast filled vessels free of background details and blurred the non-opacified cardiac outline. This mode displays the differences between the two successive images (i.e., frame 2 minus frame 1, frame 3 minus frame 2, and so on). The image processor is the heart of DPSA because it is the part of the system where subtraction and image enhancement takes place. A general flow of information in the DPSA system is shown in Figure 1.

## 3. Results

The patients’ demographics and clinical data, including the gestational age, echocardiogram findings, treatment of PH, and hemodynamic data, are summarized in the Table 1. Because of their chronic lung disease due to prematurity and continued need for respiratory support, pulmonary hypertension was diagnosed in all three patients, after birth, based on an echocardiogram. All children had normal renal function and no adverse effects of contrast after the DPSA was analyzed.

We demonstrated significant lung perfusion defects using DSPA in all three patients and compared those results with DSPA images of a normal child with normal pulmonary artery pressure (Figure 2). Patients 1 and 2 had more segmental perfusion defects than patient 3. The mean pulmonary artery pressure to mean aortic pressure ratios in patients 1, 2, and 3 were 54%, 55%, and 43%, respectively. Similarly, the pulmonary vascular resistance index to systemic vascular resistance index ratios in patients 1, 2, and 3 were 0.36, 0.31, and 0.25, respectively (Table 1). Although, these are only three patients, the preliminary data suggest that a severe perfusion abnormality correlates to the severity of pulmonary vascular disease in children with BPD. However, larger studies will be necessary to draw final conclusions. 

## 4. Discussion

Pulmonary hypertension is commonly associated with BPD, affecting one in six premature infants, and persisting to discharge in most survivors [6]. As premature infants are affected by fetal growth restriction, alterations of the pulmonary vasculature are commonly associated with BPD [7]. Premature infants should have an echocardiogram to screen for PH if there is continued need for ventilator support at postnatal day seven, as echocardiogram evidence of PH at day seven suggests a high risk for BPD and may alter therapy [8]. The perfusion defects demonstrated by DPSA in our patients may represent dysmorphic or underdeveloped pulmonary vasculature, which have been described by histopathology in infants with BPD [9].

Patients with BPD can have varied clinical manifestations or phenotypes, including lung parenchymal disease, pulmonary vascular disease, and airway disease. It is important to know whether certain segments of the vascular tree are preferentially altered with pulmonary hypertensive changes in BPD, as this will help appropriate ventilation management and prevent worsening the ventilation/perfusion mismatch. A lung biopsy and histology can evaluate microvascular changes, but this is an invasive surgical procedure and entails increased risk. Imaging of pulmonary vasculature can be obtained by standard angiography, but its definition can be limited due to overlap and parallax. Imaging of pulmonary vasculature can be obtained by a 3-D data set and stereology, but as the pulmonary capillary network functions according to the sheet flow principle, the implementation of stereological methods is made challenging [10].

The principal advantages of DSPA include a low-contrast load, lower radiation exposure, and a possible decrease in the risk of precipitation of pulmonary hypertensive crises. Other possible advantages may include a greater sensitivity than standard angiography in demonstrating segmental vascular filling abnormalities, and a higher temporal and spatial resolution, to better delineate the pre-capillary arterioles and venules during the venous phase. The principal disadvantage in our technique is the need for controlled apnea during the acquisition of DSPA images. With further improvement in the technology, we hope to mask respiratory movements and may not need to suspend respiration while obtaining a pulmonary angiogram. Digital subtraction angiography is now less routinely done and is being replaced by computed tomography angiography (CTA) and magnetic resonance angiography (MRA), which can produce 3D images. The advantages and cost-effectiveness of our method need to be compared with modern imaging modalities like 3D reconstructed images of the pulmonary vasculature obtained by CTA and MRA. 

In summary, DSPA can be a potentially useful imaging modality to assess changes in pulmonary vasculature in children who have PH associated with BPD. Our findings, if substantiated in larger studies, may shed light on the pathophysiological significance of the alteration of pulmonary vasculature in ex-premature children with BPD and aid in management and prognostication.

## Figures and Tables

**Figure 1 medsci-07-00026-f001:**
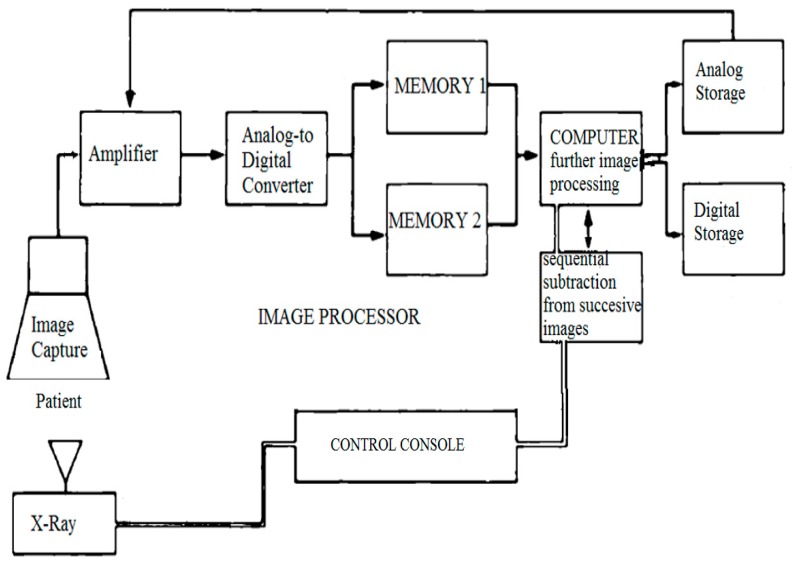
Flow of information in a generalized digital subtraction system.

**Figure 2 medsci-07-00026-f002:**
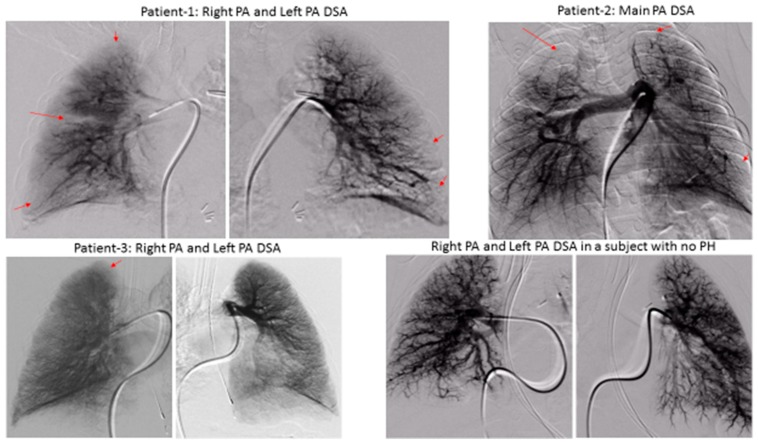
Digital Subtraction Pulmonary Angiography showing segmental perfusion defects (capillary phase, pointed by arrows) in three patients versus a normal subject with no pulmonary hypertension. (PA: pulmonary artery, DSA: digital subtraction angiography, PH: pulmonary hypertension).

**Table 1 medsci-07-00026-t001:** Summary of Clinical and Hemodynamic Data.

Patients	Echocardiogram Prior to Cardiac Catheterization	PH Medications Prior to Cardiac Catheterization	Cardiac Hemodynamics at the Time of DSPA with Baseline O_2_
#1: 4 months old female, wt:3 kg, BSA: 0.18 m^2^ at cardiac cath, 22 weeks gestation at birth, chronic lung disease, h/o large PDA, s/p closure using a 7.5 mm muscular ventricular septal defect closure device at 2 weeks, required ventilator for 1.5 months after birth	RV dilatation and reduced function, RV systolic pressure based on TR: 36 mmHg plus RAP	Oxygen by NC 2 L/min, Milrinone @0.5 mcg/kg/min Sildenafil 1 mg/kg PO tid Bosentan 4 mg/kg PO bid	PA pressure: 38/15, m 24 mmHg Ao Pressure: 59/35, m 44 mmHg RA: 10 mmHg PCWP: 10 mmHg PVRi: 1.87 WU. m^2^ SVRi: 5.16 WU. m^2^ PVRi/SVRi: 0.36
#2: 2 years old female, wt: 14.1 kg, BSA: 0.58 m^2^ at the time of cardiac cath, 22 weeks gestation at birth, chronic lung disease, s/p tracheostomy, required ventilator for 4 months after birth	PFO with right to left shunting, RV dilatation, RV systolic pressure based on TR: 52 mmHg plus RAP	Room Air Sildenafil 1 mg/kg PO tid Bosentan 4 mg/kg PO bid	PA pressure: 48/28, m 38 mmHg Ao Pressure: 90/52, m 69 mmHg RA: 10 mmHg PCWP: 8 mmHg PVRi: 7.1 WU. m^2^ SVRi: 23.08 WU. m^2^ PVRi/SVRi: 0.31
#3: 9 months old male, wt: 6.4 kg, BSA: 0.3 m^2^ at the time of cardiac cath, 24 weeks gestation at birth, chronic lung disease, h/o large PDA closure after use of ibuprofen after birth, required ventilator for 2 months after birth	RV dilatation, RV systolic pressure based on TR: 44 mmHg plus RAP	Oxygen by NC 2L/min Sildenafil 1 mg/kg PO tid Bosentan 4 mg/kg PO bid	PA pressure: 36/15, m 25 mmHg Ao Pressure: 73/44, m 58 mmHg RA: 7 mmHg PCWP: 10 mmHg PVRi: 3.3 WU. m^2^ SVRi: 13.45 WU. m^2^ PVRi/SVRi: 0.25

(wt: weight, BSA: body surface area, h/o: history of, PDA: patent ductus arteriosus, s/p: status post, cath: catheterization, NC: nasal cannula, bid: twice daily, PO: by mouth, tid: three times daily, PFO: patent foramen ovale, RV: right ventricle, RA: right atrium, TR: tricuspid regurgitation, RAP: right atrial pressure, PA: pulmonary artery, m = mean, Ao: aorta, PCWP: pulmonary capillary wedge pressure, PVRi: pulmonary vascular resistance index, SVRi: systemic vascular resistance index, WU: Woods Units).

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
