# Peer review of "Digital Subtraction Pulmonary Angiography in Children with Pulmonary Hypertension due to Bronchopulmonary Dysplasia"

_medsci, 2019, doi:10.3390/medsci7020026_

Round 1

Reviewer 1 Report

 The authors suggest that DSPA may be useful in evaluating the severity of pulmonary vascular disease in children with BPD. There report is well written. I have a few questions.

 1) DSPA may be a useful in assessing pulmonary hypertension, the authors did not compare DSPA to MRI. It is important to know whether DSPA is better in assessing PH in BPD compared to MRI, given the risk of irradiation with DSPA.

2) Given the reported incidence of BPD, I wonder why the authors assessed only 3 patients for such as important diagnostic tool?

3) Detailed description of the computer algorythm is necessary for publication.

Minor suggestion- exposure to hypoxia and not hypoxemia, in the first two weeks can induce PH    

Author Response

Response to Reviewer-1 comments

The authors suggest that DSPA may be useful in evaluating the severity of pulmonary vascular disease in children with BPD. There report is well written. I have a few questions.

DSPA may be a useful in assessing pulmonary hypertension, the authors did not compare DSPA to MRI. It is important to know whether DSPA is better in assessing PH in BPD compared to MRI, given the risk of irradiation with DSPA.

Thank you for the comment. We did not compare the DPSA findings with MRI. There is no study available in the literature to draw conclusion. This is our pilot feasibility study. We plan to compare DPSA with MRA of pulmonary vasculature. At this time, this is beyond the scope of this paper.

Rapid whole chest MRA using parallel imagining is helpful, especially there is no need of controlled apnea and the scanning time is extremely short. However, even with MRA using parallel imaging,  the capillary phase and the levo (venous) phases are still difficult to obtain, as lamellar blood flow at very slow speed (which occur physiologically at the alveolar level) produces poor signals.

Given the reported incidence of BPD, I wonder why the authors assessed only 3 patients for such as important diagnostic tool?

This is our preliminary data. As reviewer pointed out in previous comment, risk of radiation exposure with this technique, we have not adopted this universally. We are planning to do this in all our patients in future in a prospective manner, after financial assistance from a grant support.

Detailed description of the computer algorithm is necessary for publication.

Thank you. The details of physics behind the digital subtraction using the GE Healthcare’s software beyond the scope of this paper. However, we added a generalized flow plan of DSA to the revised paper (Figure-1).

Minor suggestion- exposure to hypoxia and not hypoxemia, in the first two weeks can induce PH    - The corrections are done.

Reviewer 2 Report

The paper do not covers briefly the randomized trials in the literature related to the topic. Which kind of equipment they used?

It is more a CASE REPORT than an original article.

There are only 3 cases and the power of the study is low.

There are no discussions about the renal risk.

DSA is done less at this moment in imaging departments and replaced by 3D images technique (resonance angiography) which avoid nefrotoxic agents.

Author Response

The paper do not covers briefly the randomized trials in the literature related to the topic. Which kind of equipment they used? It is more a CASE REPORT than an original article. There are only 3 cases and the power of the study is low.

Thank you. We agree that this is a brief report on only 3 cases for our preliminary experience on DSPA. The conclusion from this study needs to be validated from a large cohort with adequate power. We used the GE Healthcare’s AngioViz Vascular vision for our DSPA. We do not want to quote this commercial software in the paper, as DPSA can be obtained using other equipment. We discussed the basic principles behind the DPSA and used a generalized flow chart (Figure-1) for demonstration.

There are no discussions about the renal risk.

Thank you. We added a line stating because of our small contrast agent use, no adverse effect on renal function in all 3 cases.

DSA is done less at this moment in imaging departments and replaced by 3D images technique (resonance angiography) which avoid nephrotoxic agents.

We agree with the comment by reviewer that 3-D and 4-D images may offer reconstruction of pulmonary vascular tree. In very few places in Europe, synchrotron-based X-ray tomography has yielded interesting results about airway and vascular lung development (Ref: Ackerman et al. Angiology 2014; 14:541-51), but is not widely available. We described this point in the discussion section, that DPSA is less conventionally used recently. In this small series, we are focusing to demonstrate a feasibility of DPSA study to show the morphological vascular disease in PH associated with BPD in extremely premature infants. Future work on this will compare the time and cost required for 3-D processing vs our DPSA method.

Round 2

Reviewer 1 Report

I have no comments